# Defying decomposition: the curious case of choline chloride

Adriaan van den Bruinhorst ®[1] ✉, Jocasta Avila ®[1], Martin Rosenthal ®[2], Ange Pellegrino ®[1], Manfred Burghammer ®[3] & Margarida Costa Gomes ®[1] ✉

Chemists aim to meet modern sustainability, health, and safety requirements by replacing conventional solvents with deep eutectic solvents (DESs). Through large melting point depressions, DESs may incorporate renewable solids in task-specific liquids. Yet, DES design is complicated by complex molecular interactions and a lack of comprehensive property databases. Even measuring pure component melting properties can be challenging, due to decomposition before melting. Here we overcame the decomposition of *the* quintessential DES constituent, choline chloride (ChCl). We measured its enthalpy of fusion ($13.8 \pm 3.0$ kJ·mol) and melting point ($687 \pm 9$ K) by fast scanning calorimetry combined with micro-XRD and high-speed optical microscopy. Our thermodynamically coherent fusion properties identify ChCl as an ionic plastic crystal and demonstrate negative deviations from ideal mixing for ChCl—contradicting previous assumptions. We hypothesise that the plastic crystal nature of ammonium salts governs their resilience to melting; pure or mixed. We show that DESs based on ionic plastic crystals can profit from (1) a low enthalpy of fusion and (2) favourable mixing. Both depress the melting point and can be altered through ion selection. Ionic plastic crystal-based DESs thus offer a platform for task-specific liquids at a broad range of temperatures and compositions.

When is my deep eutectic solvent (DES) liquid? The answer can be read in a solid−liquid equilibrium phase diagram that shows the temperatures and compositions for which a mixture is liquid, solid, or solid + liquid (Fig. 1). Liquid mixtures containing choline chloride (ChCl)—the most studied DES salt—are currently explored as electrolyte media for versatile applications such as electrodeposition, pharmaceuticals, nanomaterial synthesis, and batteries;[1,2] replacing conventional solvents to meet modern sustainability, health, and safety requirements[1,3,4]. Accurate ChCl melting properties are essential to predict and describe whether ChCl-based mixtures remain liquid at the operating temperature of the application[5–7]. Unfortunately, ChCl decomposes before it melts when heated at typical rates ($<1$ K·s$^{-1}$)[5,8,9]. Its melting point ($T_{fus}$) and molar enthalpy of fusion

($\Delta_{fus}H_m$) can therefore not be determined using conventional calorimetric techniques.

A wide range of $\Delta_{fus}H_m$ values were estimated for ChCl[5,10,11]. Figure 1 shows the impact of this discrepancy on the predicted phase diagram—and thus the boundaries of application—for a model mixture containing ChCl. The scatter originates in (i) the different nature of the data used for the $\Delta_{fus}H_m$ estimation and (ii) the necessary assumptions about the thermodynamics of mixing (see Supplementary Table 5). The currently most widely accepted ChCl fusion properties are $\Delta_{fus}H_m = 4.3$ kJ·mol$^{-1}$ and $T_{fus} = 597$ K[5], but these values are thermodynamically inconsistent with the carefully measured phase diagrams by Alhadid et al.[8], from which higher values would be expected. To address these contradictory

[1]École Normale Supérieure de Lyon and CNRS, Laboratoire de Chimie, Ionic Liquids Group, 46 allée d'Italie, 69364 Lyon Cedex 7, France. [2]Department of Chemistry, KU Leuven, Celestijnenlaan 200F, Box 2404, 3001 Leuven, Belgium. [3]ESRF, The European Synchrotron, 71 Avenue des Martyrs, CS40220, 38043 Grenoble Cedex 9, France. ✉e-mail: adriaan.van-den-bruinhorst@ens-lyon.fr; margarida.costa-gomes@ens-lyon.fr

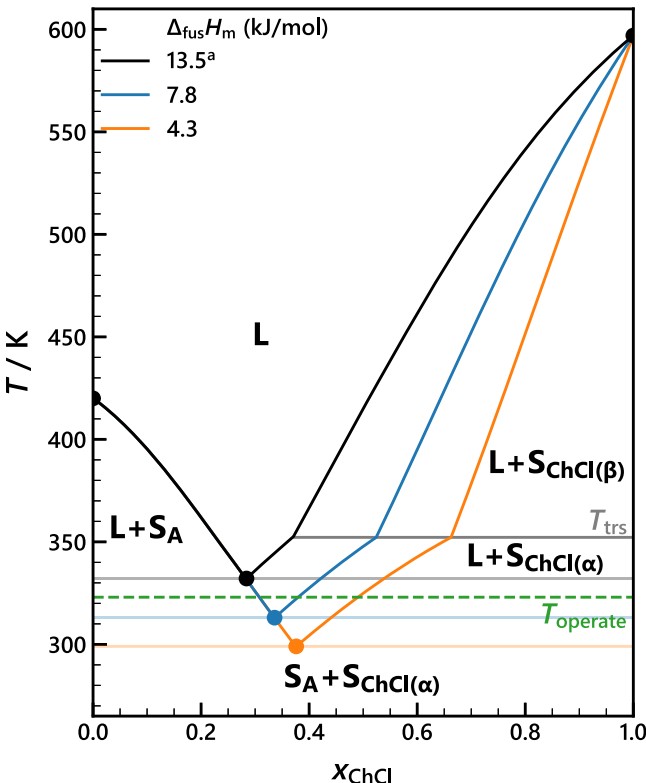

**Fig. 1 | Phase diagrams from different molar enthalpies of fusion reported for choline chloride.** The impact of the variety in reported choline chloride (ChCl) molar enthalpies of fusion ($\Delta_{fus}H_m$ = 4.3[5], 7.8[10], 13.5[a] [11] kJ · mol⁻¹ [11], see Supplementary Table 5) on the phase diagram of regular solution A + ChCl assuming $\chi$ = − 2.5[32] and omitting $\Delta C_{P,m}$. The ChCl $\alpha \rightarrow \beta$ solid–solid transition temperature ($T_{trs}$) and operating temperature are highlighted in grey. [a]Corrected for $\Delta_{trs}H_m$.

findings and solve the long-standing issue of inaccurate ChCl fusion properties, we went beyond the experimental conditions of conventional calorimetric techniques to gain direct experimental access to the melting transition of ChCl.

Inspired by recent studies towards decomposing biomolecules[12,13], we defied ChCl decomposition by using (ultra-)fast differential scanning calorimetry (FDSC) to heat ChCl rapidly, to temperatures where the melting kinetics surpass degradation (Supplementary Fig. 16). The calorimeter-on-a-chip design of FDSC[14,15] grants access to heating rates up to 50 000 K · s⁻¹, because the thermal lag of the furnace and microscopic samples (1 ng to 100 ng) is substantially reduced as compared to regular DSC. Since ChCl is strongly hygroscopic, we adapted the experimental setup such that all samples were prepared and measured under dry argon or nitrogen atmosphere. A full description of the experimental setups and (tedious) sample preparation for ChCl is given in the Supplementary Methods.

With FDSC, we experimentally assessed the phase transitions of ChCl at extreme heating rates. The underlying structural processes of these transitions cannot be identified based on enthalpic data alone, so direct access to structural information is needed. We therefore combined fast scanning calorimetry (FSC) with synchrotron micro-XRD, which allowed us to follow the evolution of the crystal structure before and after liquefaction of ChCl. The unique FSC + micro-XRD setup at beamline ID13 of the European Synchrotron (ESRF)[16–19] collects a full scattering pattern every 2 ms for ng-sized samples under dry and inert atmosphere, giving an excellent temperature resolution at these extreme heating rates. We also combined FSC with high-speed optical microscopy to evaluate the sample morphology upon heating and cooling. We could then identify the various thermal events obtained by FDSC.

## Results and discussion

As expected, ChCl completely decomposed at rates < 1000 K · s⁻¹; decomposition was characterised by noisy endothermic peaks in FDSC (Supplementary Fig. 18). Only a dark residue remained after heating ChCl until 730 K at 100 K · s⁻¹ (Supplementary Fig. 25). However, from heating rates of 1000 K · s⁻¹ and upwards, the FDSC thermograms became less noisy and more distinct thermal events can be recognised (Supplementary Fig. 21). Figure 2A shows the heat flow signal of a ChCl particle heated at 5000 K · s⁻¹ with two distinct endothermic peaks. We attribute the first sharp peak to melting, the second broad and relatively noisy peak is typical for decomposition-related processes[20]. The inset shows the well-established $\alpha \rightarrow \beta$ solid–solid transition of ChCl[9] before and after melting. The re-appearance of the transition confirms that ChCl recrystallises in its original crystal lattice upon cooling. The reduced $\alpha \rightarrow \beta$ transition enthalpy and sample baseline–both directly proportional to the sample mass–indicate a significant mass loss due to partial decomposition.

To the best of our knowledge, we present the first evidence of reversible melting for ChCl based on FSC combined with micro-XRD measurements. Figure 2C shows the evolution of ChCl XRD patterns with temperature when heated at 1000 K · s⁻¹ and cooled subsequently as rapidly as possible. Upon heating, we could clearly observe the $\alpha \rightarrow \beta$ transition, as well as the thermal expansion of $\beta$-ChCl signified by peak-shifts to slightly lower q at higher temperatures. Around 730 K, the XRD pattern shows an amorphous halo and an absence of crystallographic reflections, indicating the complete liquefaction of ChCl. This corresponds to the final temperature of the first peak in Fig. 2A, reinforcing that this peak can be attributed to melting. Upon cooling, $\beta$-ChCl peaks re-appeared at their original position. At rates ≥1000 K · s⁻¹, the melting and subsequent recrystallisation were reproducible (Supplementary Movies 1–8, Supplementary Table 3).

Also visually we could clearly observe the formation of liquid at $T$ > 700 K using FSC combined with high-speed microscopy imaging (Fig. 2D, Supplementary Movies 9–12, Supplementary Table 4). The liquefaction was almost immediately followed by some gas release and a simultaneous sample size reduction. Visual inspection of the sample before and after melting and recrystallisation + micro-XRD showed that the samples became smaller after each heating–cooling cycle (Supplementary Fig. 26). The mass loss and gas formation can be assigned to partial decomposition shortly after melting, or to the direct evaporation of liquid ChCl as was observed with FDSC for low vapour pressure ionic liquids[14].

ChCl melts at a temperature of 687 ± 9 K, as measured from the onset of the first FDSC peak at 5000 K · s⁻¹ (Fig. 3). We obtained $\Delta_{fus}H_m$ from the integral of the same peak, using the integrated $\alpha \rightarrow \beta$ transition peak ($\Delta_{trs}H$) and $\Delta_{trs}H_m$ from literature (16.3 kJ · mol⁻¹)[9] as internal reference. The melting and decomposition peaks overlapped to some extent for all experiments. We therefore integrated the first peak using two approaches: (i) fitting all thermal events to a melting + decomposition model (ii) fitting a single peak to the data until the inflection point after the melting peak (Fig. 3, Supplementary Methods: Fast differential scanning calorimetry). The two approaches yielded similar $\Delta_{fus}H_m$ at $T_{fus}$ and normal pressure: 13.2 ± 4.1 kJ · mol⁻¹ and 13.8 ± 3.0 kJ · mol⁻¹, respectively. Approach (ii) was adopted as it requires no assumptions towards the nature of the second peak and yields a lower statistical error. The corresponding molar entropy of fusion is 20.2 ± 4.4 J · mol⁻¹ · K⁻¹ ($\Delta_{fus}S_m = \Delta_{fus}H_m/T_{fus}$). Individual values, FDSC thermograms, and a detailed statistical analysis are reported in Supplementary Tables 1–2 and Supplementary Figs. 19–24. To the best of our knowledge, this is the first direct experimental measurement of $T_{fus}$ and $\Delta_{fus}H_m$ for ChCl.

Our results allow for the quantitative classification of $\beta$-ChCl as an ionic plastic crystal[8,21]: it has a low molar entropy of fusion (20.2 J · mol⁻¹ · K⁻¹), a high molar entropy of S–S transition (46.3 J · mol⁻¹ · K⁻¹)[9], and a disordered face-centred cubic crystal lattice

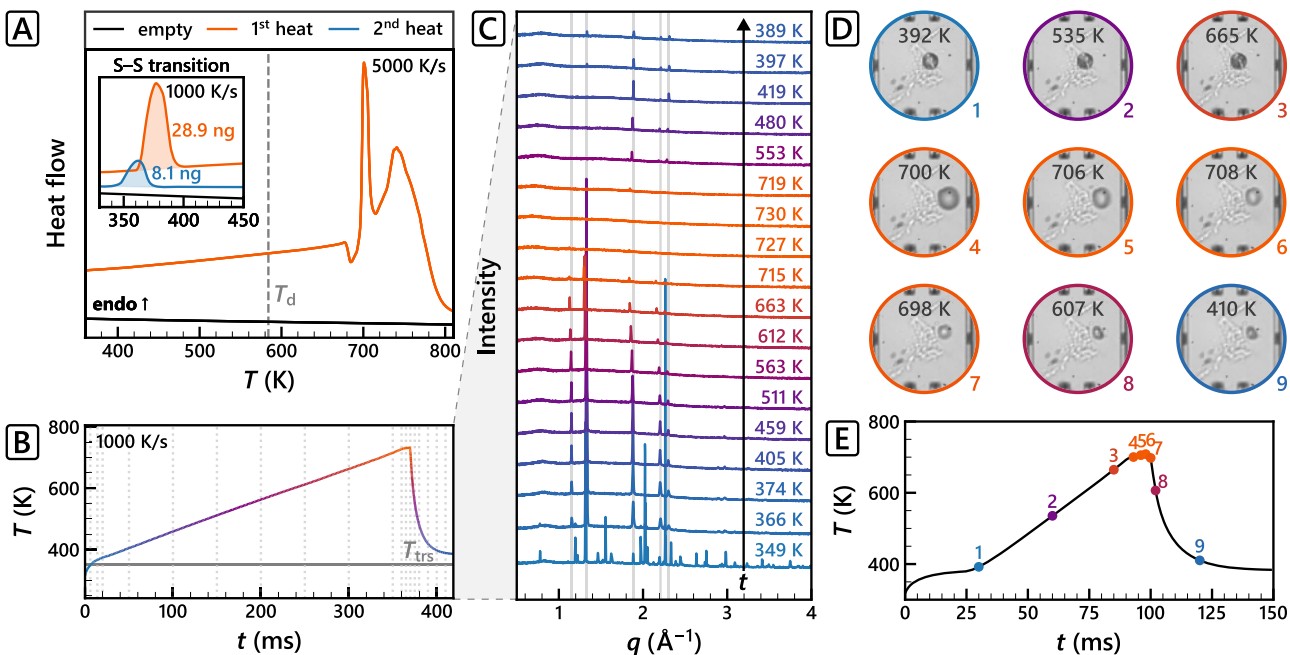

**Fig. 2 | Fast scanning calorimetry of choline chloride proving melting before decomposition and recrystallisation at heating and cooling rates of 1000 K · s⁻¹ and more. A** Heat flow signal of choline chloride (ChCl) obtained with fast differential scanning calorimetry at a heating rate of 5000 K · s⁻¹ and 1000 K · s⁻¹ (inset). The decomposition temperature ($T_d$) is highlighted in grey. The sample mass in the inset was determined from the integral of the ChCl solid-solid (S–S) $\alpha \rightarrow \beta$ transition and the corresponding molar enthalpy (16.3 kJ · mol⁻¹)[9]. **B** Temperature ($T$) vs. time ($t$) profile of ChCl using a fast scanning calorimeter at a heating rate of 1000 K · s⁻¹. The ChCl S–S transition temperature ($T_{trs}$) is highlighted in grey. **C** Corresponding XRD patterns at selected temperatures, which are highlighted as vertical grey dotted lines in the $T−t$ profile and increase in $t$ from bottom to top (black arrow). Also highlighted are the $q$ of $\beta$-ChCl. **D** Selection of high-speed microscopy images of a choline chloride particle on silicon grease heated at 5000 K · s⁻¹ and **E** the corresponding $T−t$ profile, where $T$ and $t$ are highlighted for each image.

above $T_{trs}$[22]. The molar entropy of fusion of ChCl is significantly higher than that of other choline-based plastic crystals with bulkier anions ([BF₄]⁻,[ClO₄]⁻, and [H₂PO₄]⁻, see Supplementary Table 6). NMR studies show that the choline cation gains significant isotropy upon (or just before) the solid−solid transition (Supplementary Table 6). Contrary to the entropy of fusion, the entropy of transition towards the plastic crystal state is similar for the different choline salts (Supplementary Table 6). This indicates that the residual entropy in the crystal is largely dictated by the anion, offering a design parameter to the enthalpy of fusion and thus the extent of the (ideal) melting point depression when forming choline-based DESs.

None of the previously estimated fusion properties equal our results. Only after correcting the estimate of $\Delta_{fus}H_m = 29.8$ kJ · mol⁻¹ [11] with $\Delta_{trs}H_m$ (Supplementary Table 5), a matching value of 13.5 kJ · mol⁻¹ could be calculated. The currently widely accepted fusion properties[5]– $T_{fus} = 597 \pm 7$ K and $\Delta_{fus}H_m = 4.3 \pm 0.6$ kJ · mol⁻¹—are significantly lower than our accurate FDSC results. While the accuracy of $T_{fus}$ could be improved by extrapolating to 0 K · s⁻¹, FDSC results at different heating rates imply that this effect is small as compared to the large difference with the previously estimated $T_{fus}$ (Supplementary Fig. 22). Fernandez et al.[5] estimated their fusion properties from optically determined liquefaction temperatures while heating ChCl-rich mixtures at 8.33 mK · s⁻¹. Similarly, Silva et al.[23] reported melting point depressions for urea + ChCl at 1.67 mK · s⁻¹. Using thermogravimetric analysis we show that ChCl-rich mixtures are not thermally stable under those conditions (Supplementary Fig. 17). Corresponding liquefaction temperatures should therefore not be interpreted as SLE data.

To circumvent decomposition and to gain direct access to the liquidus temperatures, we explored ChCl-rich mixtures of urea + ChCl ($x_{ChCl} = 0.7, 0.8, 0.9$) with FSC + micro-XRD (Supplementary Movies 1−8, Supplementary Table 3). As expected, all components solidified below the eutectic point ($x_e$, $T_e$) resulting in scattering patterns for urea and ChCl. Above $T_e$, urea and part of the ChCl melted, and above $T_{trs}$ we

confirmed that the excess solid is pure $\beta$-ChCl. But−similarly to others[8,24]−we did not observe clear melting point depressions at ChCl-rich compositions. Instead, the onset of the melting process remained virtually constant with composition (Fig. 4). This does not correspond to thermodynamic equilibrium and implies a superheated metastable solid phase.

The resilience to melting of $\beta$-ChCl was also observed for other tetraalkylammonium-based salts[25,26]. This behaviour might be explained by the plastic crystal nature of these salts. Upon heating, thermal energy is not absorbed by the crystal lattice but is instead absorbed by increased orientational motion of the ions[21,27,28]. Our hypothesis is therefore that the thermal energy is kinetically trapped when heating salt-rich mixtures. The pure plastic crystal phase is thus superheated, which prevents its equilibrium with the liquid mixture. Upon slow heating, however, partial thermal decomposition allows the thermal energy to be absorbed by the weakened lattice rather than the isotropic ions within the lattice. The result is a liquefaction temperature that is limiting towards the decomposition temperature instead of $T_{fus}$ (Fig. 4). Although beyond the scope of this study, this hypothesis could be verified spectroscopically by evaluating the orientational motion of the ions as a function of temperature for mixtures with excess solid salt[27].

Curiously, $\beta$-ChCl did not readily recrystallise into the $\alpha$-phase at temperatures well below $T_{trs}$, unless touched with a microscopic manipulation probe. The physical impact on the ng-sized particle was probably enough to overcome the energy barrier of the plastic crystal lattice at ambient temperature. When $\beta$-ChCl would crystallise upon cooling, the S−S peak would sometimes split on heating (Supplementary Fig. 11). The peaks would reunite at $T_{trs}$ after a prolonged isotherm. The reversibility of the peak splitting suggests the formation of metastable solid phases. After extensive method optimisation (Supplementary Discussion: S−S transition of ChCl) we could recrystallise $\beta$-ChCl without physical impact. The sample mass calculated from the resulting $\Delta_{trs}H$ matched the mass derived from heat capacity

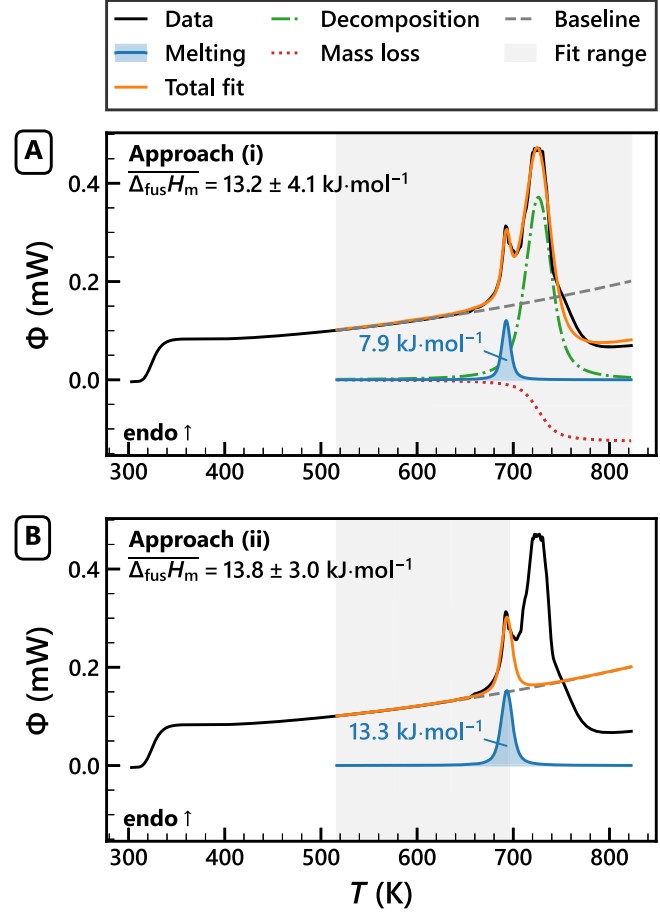

**Fig. 3 | Two approaches to fit the choline chloride melting peak.** Heat flow ($\Phi$) signal of a choline chloride particle (black line, no. 10 in Supplementary Table 1). The data was fit using **A** model (i) melting + decomposition, or **B** model (ii) a single Voigt profile (Supplementary Methods: Fast differential scanning calorimetry). Shown are the mean molar enthalpy of fusion ($\overline{\Delta_{fus}H_m}$), the model fit (orange line), the fitted data range (grey area), as well as the components of each model: quadratic baseline (grey dashes), the melting peak and $\Delta_{fus}H_m$ (blue line and area), the decomposition peak (green dash-dotted line), and mass loss contribution (red dotted line).

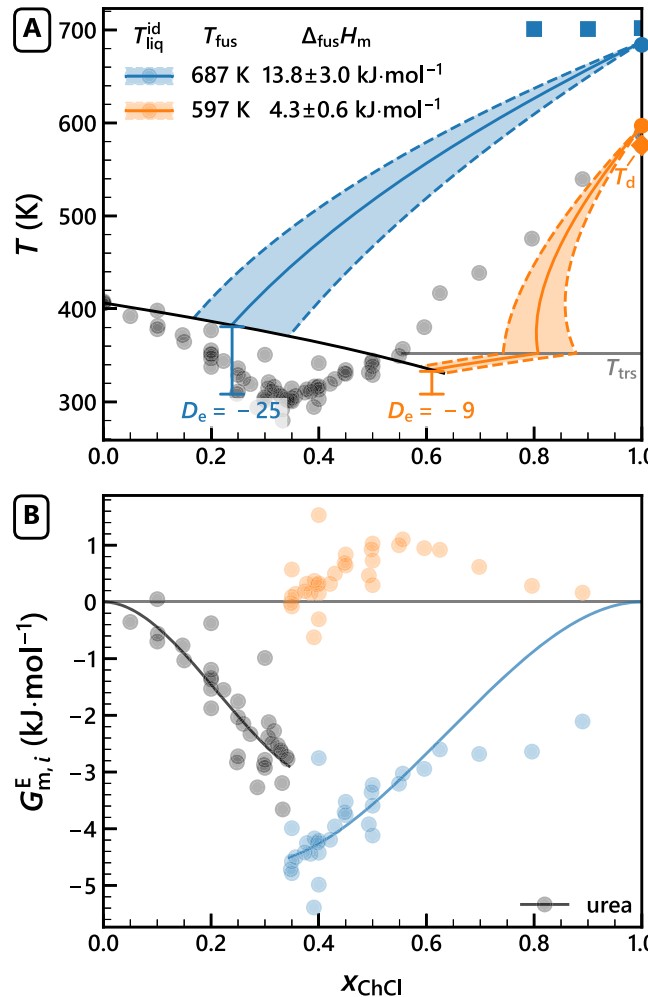

**Fig. 4 | Ideal melting point depression and excess Gibbs energy of choline chloride using the extrapolated and the directly measured fusion properties.** **A** Literature liquidus temperatures for urea + choline chloride[31] (black circles) with the ideal liquidus temperature ($T_{liq}^{id}$, solid lines) calculated using different values for the ChCl enthalpy of fusion ($\Delta_{fus}H_m$) and their 95% confidence interval (dashed lines), taking into account $\Delta_{fus}C_{P,m,\,ChCl} = 19.3\,\mathrm{J\cdot mol^{-1}\cdot K^{-1}}$ and $\Delta_{trs}C_{P,m,\,ChCl} = 20\,\mathrm{J\cdot mol^{-1}\cdot K^{-1}}$[19] (Supplementary Discussion: Impact of fusion properties). The eutectic depth ($D_e$) is shown for both $\Delta_{fus}H_m$ values. **B** Partial molar excess Gibbs energy ($G_{m,i}^E$, circles) derived from the experimental liquidus temperatures and the ideal liquidus temperatures shown in (**A**). Solid lines highlight the general trend of $G_{m,i}^E$ from first-order Redlich–Kister fits. Supplementary Table 7 lists literature data and calculated $G_{m,i}^E$.

remarkably well (Supplementary Discussion: Sample mass determination), confirming that $\Delta_{trs}H$ is a suitable internal reference to calculate $\Delta_{fus}H_m$.

The accurate experimental ChCl fusion properties obtained by FDSC are imperative for a correct thermodynamic interpretation of the eutectic phase behaviour of ChCl-based mixtures (see also Supplementary Discussion: Impact of fusion properties). For the extensively studied urea + ChCl mixture, we show that both components exhibit a favourable (negative) partial molar excess Gibbs energy (Fig. 4), generating a deep eutectic with $D_e = -25$. Thermal decomposition of ChCl-rich samples was corroborated by a deviation from the general trend of the partial molar excess Gibbs energy. As shown in Fig. 4 (orange circles), inaccurate estimations of fusion properties can lead to the appearance of strong asymmetric deviations from ideality of ChCl in DESs, as hypothesised in previous studies[5,6,29]. Our experimental findings invalidate this hypothesis.

We are currently expanding our research to other key DES constituents that decompose before or upon melting, such as short-chain tetraalkylammonium halides and trimethylglycine (betaine). Many of these show plastic crystal behaviour, having high-entropy solid–solid transitions[25]. Our findings for ChCl serve as a foundation for the development of DESs centred around ionic plastic crystals. We

propose to prepare liquids from ionic plastic crystals that simultaneously profit from (1) a low $\Delta_{fus}H_m$[30] and (2) favourable mixing of the ionic and molecular compounds balancing intermolecular interactions and disorder[31]. Both aspects can be tuned by selecting the appropriate ions and molecular compounds and significantly increase the melting point depression of the salt, yielding dense ionic fluids at accessible temperatures over a broad range of compositions. Ionic plastic crystal-based DESs thus provide a platform to include solids with a low environmental or safety impact in a renewable liquid solvent with task-specific properties.

## Methods
### Chemicals and sample handling
Choline chloride (ChCl) was purchased from Acros (product purity 99%, water content of 0.7%, batch purity 100.1% by argentometric titration on a dry basis). It was recrystallised from technical grade

absolute ethanol ($w_{ChCl} \approx 0.40$ at 333 K, cooled to 301 K overnight), washed with refrigerated ethanol, dried for at least 72 h under vacuum (< 0.3 mbar) and stirring, and finally stored under dry argon atmosphere. Urea was purchased from Sigma-Aldrich (purity > 99.5%) and used as is. Urea + ChCl mixtures were prepared by weighing the appropriate amounts of each constituent to a total mass of 0.5 g. The solids were then ground together with a mortar and pestle until a homogeneous solid mixture/paste was obtained. All samples were handled under dry and inert atmosphere at all times, see Supplementary Methods for details on the glovebox and glovebags applied.

### Fast differential scanning calorimetry

A Mettler-Toledo Flash-DSC 2+ equipped with a Leica Microsystems stereo-microscope (model Leica LED3000 RL, 58 mm) with a total magnification of 40x was used for the Fast Differential Scanning Calorimetry (FDSC) measurements. The sensors were conditioned and corrected according to the procedure defined by the manufacturer. The sample stage temperature was set to 303.15 K and the reference side of the chip was left empty.

The deposited samples are in the order of 1 ng to 50 ng. To handle such small samples we used hairs or a thin metal wire installed on the pen. A thin film of highly viscous Korasilon silicon grease was employed prior to the placement of ChCl in order to improve the thermal and physical contact between the active sensor area and the ChCl particles. The grease did not show any thermal events within the temperature range and at the heating rates under study. A small crystal (1 mm$^3$ suffices) of ChCl was ground on a microscope glass slide using a small pestle to obtain particles of the appropriate size. The particle was selected by eye, picked up using a hair tip that was slightly wetted with silicon grease, and deposited to the centre of the sensor.

Typically, ChCl was exposed to two different temperature programmes: (i) a heating–cooling cycle from ambient temperature to a temperature well below $T_d$ at $1000 \, K \cdot s^{-1}$ to obtain $\Delta_{trs}H$, and (ii) a high-temperature cycle well above $T_d$ at the heating/cooling rates listed in Supplementary Table 1 without high-temperature isotherm. In between each consecutive run on the same ChCl particle, the following temperature programme was repeated five times to ensure complete recrystallisation: ChCl was heated to 343.15 K at $1000 \, K \cdot s^{-1}$ (about 10 K below $T_{trs}$), kept isothermally for 60 s, and cooled to 303.15 K at $1000 \, K \cdot s^{-1}$.

The heat flow signals were recorded using the Star-e software package, they were then integrated and visualised using Python 3. $\Delta_{trs}H$ was calculated from the integral of the solid–solid peak, which was numerically integrated using a sigmoidal baseline that scales with the peak integral. The start and end of the peak were determined by the point at which a rolling linear fit from the inflection point towards the baseline merges with the baseline. $\Delta_{fus}H$ was determined in two ways: (i) fitting all data to a melting + decomposition model, (ii) fitting a single melting peak to the data until the inflection point after the first major peak (Fig. 3, Supplementary Methods: Fast differential scanning calorimetry).

The sensors were recovered by submerging the chips in distilled water, ethanol, and petroleum ether. Adequate removal of the grease typically required several subsequent washes with petroleum ether alternated with rubbing the sensor surface using a clean hair.

### Fast scanning calorimetry combined with micro-XRD or high-speed optical microscopy

FSC with in-situ ms time-resolved micro-XRD was measured at the ID13 beamline at the ESRF. A monochromatic X-ray beam with a photon energy of 13 keV and a photon flux of approximately $3 \times 10^{12} \, s^{-1}$ was focused by means of compound refractive Beryllium lenses to a spot size of 2.5 $\mu m$ in both directions at a location 40 mm upstream of the sample position. Possible beam-induced sample damage was mitigated by adjusting the effective beam size to match the size of the

smallest sample diameter, corresponding to approximately 15 $\mu m$. An EIGER 4M single photon counting area detector from Dectris AG (Switzerland) was employed. The detector was operated at 500 Hz, slightly below its maximum sampling frequency of 750 Hz to ensure a stable performance.

The FSC is a custom device implemented at the ID13 beamline in 2014[16,17]. The sensor chip (XEN-39392, Xensor Integration, NL) constitutes a SiN$_x$ membrane of 1 $\mu m$ thickness supported by a Si frame, the active area is 100 $\mu m$ by 100 $\mu m$. Two pairs of resistance heaters are placed on the sides of the active area: (i) main heaters to apply a temperature programme to the specimen and (ii) a secondary heaters to apply a temperature offset. The resulting temperature range is 383 K to 730 K, starting above $T_{trs}$. Six thermocouples connected in series provide a large scanning and observation field for the sample temperature. The temperature was calibrated using the onset of melting of four low melting metals: indium, tin, bismuth and zinc.

ChCl was added to the sensor similarly as for FDSC, except the sample size was typically larger. The sensor was then placed in a hermetic sample cell to maintain the inert and dry atmosphere, using highly viscous Korasilon silicon grease as seal. The windows of the cell are SiN$_x$ membranes with a thickness of 1 $\mu m$ to ensure good transmission of the XRD signal. Subsequently, it was positioned and aligned in the X-ray beam using a retractable on-axis microscope. Typically, a single XRD pattern (exposure time ≤ 20 ms) was collected before the heating run to verify the acquisition conditions as well as the sample position.

The secondary heater is switched off temporarily when uploading temperature programme to the data acquisition board of the FSC prior to the experiment. The sample temperature thus drops rapidly to ambient temperature prior to the fast heating run. A trigger scheme was implemented to ensure synchronisation of the micro-XRD data and the FSC data. The FSC triggered the activation of the fast X-ray shutter as well as the data collection of the Eiger 4M detector. As a result, the sample would only be exposed to X-rays during the actual temperature programme, reducing extensive exposure of the sample prior to and after FSC data collection.

A single temperature programme was applied for each run: (1) an isotherm of 25 ms to stabilise at 383 K before heating at controlled rates, (2) a heating ramp to the maximum temperature (~730 K) at $100, 1000, 2000, or \ 5000 \, K \cdot s^{-1}$, (3) an isotherm of 10 ms, and (4) cooling as fast as possible (uncontrolled) to prevent decomposition. XRD patterns were recorded simultaneously every 2 ms. For some samples, a picture was taken before and after heating to evaluate sample morphology changes owing to the temperature programme and/or the X-ray beam exposure. The used diffraction geometry, including the modulus of the scattering vector $q$, was calibrated using alpha-alumina. The collected two-dimensional diffraction patterns were resampled and reduced to one-dimensional scattering curves using the pyFAI package (Supplementary Methods: Fast scanning calorimetry + $\mu$-XRD).

The FSC was also combined with in-situ high-speed optical microscopy imaging. To this end, an Olympus BX50 optical microscope was equipped with a Phantom v7.3 fast CCD-camera from Vision Research US with a maximum frame rate of 15 kHz. The measurements were done using bright field transmission and reflection light illumination taking advantage of the good transparency of the SiN$_x$ membranes. To synchronise the fast heating measurements with the image acquisition, the fast CCD was triggered directly by the FSC device at the beginning of each heating run. Individual images were recorded at a fixed frame rate of 5 kHz.

### Thermogravimetric analysis

Thermogravimetric analysis (TGA) measurements were carried out on a Setaram Labsys Evo TGA under a nitrogen flow of $30 \, mL \cdot min^{-1}$.

About 10 mg was weighed into a $100\,\mu$L aluminium oxide crucible under argon atmosphere. When placing the crucible in the instrument, a short exposure to the moist atmosphere could not be avoided. The sample was therefore heated from 303 K to 393 K at $10\,\text{K}\cdot\text{min}^{-1}$ and kept isothermally for 30 min to dry the sample. Then the sample was heated from 393 K to 473 K at $10\,\text{K}\cdot\text{min}^{-1}$ and further from 473 K to 573 K at $0.1\,\text{K}\cdot\text{min}^{-1}$. Finally, the crucible was cleaned by heating from 573 K to 1773 K at $30\,\text{K}\cdot\text{min}^{-1}$.

## Data availability

All FSC + micro-XRD data recorded at the European Synchrotron Radiation Facility is publicly available as of 2025 at https://doi.org/10.15151/ESRF-ES-708312842. All numerical data obtained from the fast differential scanning calorimetry are listed in Supplementary Tables 1 and 2, and corresponding thermograms are shown in Supplementary Figs. 19–21.

## Code availability

The code used to analyse the data in this study is available upon request.

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

## Acknowledgements

The authors thank Dimitri A. Ivanov from Institut de Sciences des Matériaux de Mulhouse for providing the high-speed camera for the in-situ fast optical microscopy measurements. We thank Ahmad Alhadid for the scientific discussions on plastic crystals and ChCl-rich liquidus temperatures. We thank Nicolas Scaglione for helping with the thermogravimetric analysis and Inês Vaz for attentively evaluating the manuscript. We acknowledge IDEX-LYON for financial support

(Programme Investissements d'Avenir ANR-16-IDEX-0005). We acknowledge the European Synchrotron Radiation Facility (ESRF) for provision of synchrotron radiation facilities and we thank Michael Sztucki for assistance and support in using beamline ID13.

## Author contributions

A.v.d.B. conceived the study, designed and performed experiments, processed data, and drafted and revised the manuscript. J.A. performed experiments and reviewed data as well as the manuscript. M.R. designed and performed ESRF-based experiments, processed data, and reviewed the manuscript. A.P. performed experiments and reviewed data as well as the manuscript. M.B. supported ESRF-based experiments, processed data, and reviewed the manuscript. M.C.G. conceived the study, designed the experiments, and drafted and revised the manuscript.

## Competing interests

The authors declare no competing interests.
