## [Peer Review File · Nature Communications]

REVIEWER COMMENTS

Reviewer #1 (Remarks to the Author):

This paper describes some incredibly difficult, challenging measurements that deliver, in the end, a simple but exceptionally important results - namely, an experimentally determined measurement of the 'real' melting point and corresponding enthalpy of fusion of choline chloride. This can no be used to support theory in modelling and predicting the behaviour of choline-based deep eutectic systems based on robust experimentally determined parameters, not on extrapolated predictions. In my view, this is comparable to the ground breaking work identifying that some ionic liquids are distillable (albeit at high vacuum) that, again, enables experimental data on vapour-liquid equilibrium to be benchmarked.

As such, this is noteworthy, significant, and should, in my view, be published.

The only area of the manuscript with which I feel a little confused or in doubt over is the extension of observations that the beta-form of choline chloride is a plastic crystal phase, and that the resilience of this to transforming back through the S-S to alpha-ChCl has, in some way, a role in formation of DES accessible across a broad range of compositions. While I think that this statement is qualitatively correct, I do not really think that this is a new concept and would highlight research in the ionic liquids field, from which DES emerged, on the utility of plastic crystal forming materials (Phys. Chem. Chem. Phys., 2013,15, 1339 and J. Mater. Chem., 2010,20, 2056) for consideration.

Reviewer #2 (Remarks to the Author):

The present manuscript improves the understanding of thermodynamic and morphological properties of the archetypical deep eutectic solvent (DES) component, choline chloride (ChCl), by combining fast methods. The techniques not commonly employed for studying DES and its components are fast-scanning calorimetry, synchrotron-based micro X-ray diffraction, and high-speed optical microscopy. This combination leads to a much deeper understanding of the thermodynamics of the studied components.

The results are interesting but not groundbreaking. Besides combining the three methods, there is nothing new in the methodology.

The newly determined values for the melting point and melting enthalpy are particularly useful, but their determination leaves some open questions:

Page 4 and Fig. S16: The reported T_{fus} of 687 ± 9 K corresponds to the measured value at a heating rate of 5000 K/s. This value may be influenced by thermal lag and melting kinetics. For thermodynamic considerations, the value at heating rate zero should be applied. The slopes of the lines in Fig. S16 indicate an effect of about 10 K when going from 5000 to 0 K/s. This should be considered or at least discussed.

How was the onset temperature (T_{fus}) determined? Commonly, for first-order transitions, the extrapolated peak onset temperature is considered. Its determination requires at least a linear part of the leading flank of the melting peak; see, e.g., <https://doi.org/10.1007/BF01996802>. The extrapolated peak onset temperature should be available from the measured curves, and its determination does not require any peak separation procedure. What is the reason for the two lines in Fig. S16?

What is the reason for the ca. 20 K difference between the mean value of T_{fus} at 5000 K/s (687 K) and the value from the linear fits (about 665 K)?

The reported value $T_{fus} = (687 \pm 9)$ K seems too optimistic. It should be (687 ± 20) K or so since there are systematic deviations.

The value of $\Delta_{\text{fus}}H_m$ is determined at T_{fus} . This should be mentioned since the extrapolated value at $T = 298 \text{ K}$ is commonly employed for thermodynamic considerations.

Reviewer #3 (Remarks to the Author):

This manuscript presents results from experiments quantifying, for the first time, the fusion properties of choline chloride. In a vacuum, this may appear to be a small result but in the context of the field of Deep Eutectic Solvents (DESs), it is anything but: choline chloride typically melts before decomposing, meaning that all estimates of (non-)ideal mixing of the most common DESs are based on extrapolated data, which this manuscript essentially proves to be faulty through elegant measurements. This paper therefore represents a breakthrough and with the right framing it will interest a very wide community – not just those working in applied thermodynamics.

In this manuscript, two primary techniques were deployed, both of which are advanced and highly experimentally challenging – namely, ultra-high-speed DSC and synchrotron X-ray diffraction. From my assessment, the work, methodology and analysis has been completed competently, and the manuscript was enjoyable to read due to the quality of writing. If my series of comments, questions and recommendations can be handled, I would unreservedly support the publication of this important article in Nature Communications.

Questions and comments:

1. Regarding the framing: I believe that the authors could do more to contextualize the broad impact of the results. At present, the introduction and conclusion focus quite strongly on the context of thermodynamic treatments of DESs. For example, in the conclusion, the authors currently state: “The accurate experimental ChCl fusion properties obtained by FDSC are imperative for a correct thermodynamic interpretation of the eutectic phase behaviour of ChCl-based mixtures,” and in the introduction, the work is described as “the long-standing issue of inaccurate ChCl fusion properties”. While these statements are indeed true and fascinating, I would argue that this work has broader impact than just the above, since it also contributes to the overall definition, and applications, of the entire field of DESs. I encourage the authors to go a step further in communicating the impacts of these results more explicitly, and accordingly strengthen introductory and concluding arguments and hypotheses such as the above.
2. I think that a little bit more discussion, in the context of the literature, would go a long way towards clarifying the manuscript’s significance to a wider audience. The points raised throughout the manuscript about ionic plastic crystals is particularly interesting and evocative.
 - a. Reading the manuscript implies that this is the first discovery showing that ChCl can be categorized as an ionic plastic crystal. Is this so? If yes, it could be explicitly stated. Either way, the novelty of this result requires comparison and discussion, in the context of the known plastic crystalline nature of compounds such as ChBF₄ (10.1515/zna-1997-8-923), and prior reports of the solid-solid transitions of ChCl and related quaternary ammonium compounds (10.1063/1.436673 and 10.1016/0040-6031(70)80027-2).
 - b. The authors conclude that these “findings for ChCl serve as a foundation for the development of DESs centred around ionic plastic crystals,” and write in the abstract that these “results thus pave the way for ionic plastic crystal based DESs.” The findings make me wonder whether, in fact, using ionic plastic crystals is already the de facto design rule for most known common DES, based on for example ChCl or quaternary ammonium salts, but that this has not yet been noticed?
 - i. To take this argument one step further – if having some ionic plastic crystalline nature present in at least one of the components is in some way ‘key’ or ‘essential’ to a DES, such a result would be represented in experimental structure measurements. Elastic and inelastic neutron scattering data consistently shows disorder in the DES bulk, likely due to a low energy barrier to cross between different accessible structural configurations (i.e. as discussed in 10.1039/C5GC02914G,

10.1063/1.5010246 and 10.1039/C7CP01286A).

ii. While many other potential DES constituents decompose on or before melting, can betaine (spelt incorrectly as trymethylglycine in the concluding paragraph) be considered as such? Again, while it is not stated explicitly, the flow of the text implies this. Consider prior measurements (i.e. 10.1039/C4CP05094K).

3. The authors describe visual evidence for liquefaction, gas release, and a subsequent reduction in sample size following thermal cycling. I agree that this may well be due to vaporization of ChCl, but partial decomposition is difficult to rule out without further analysis, even if the remnant ChCl recrystallizes. Was, or could, any further quantitative analysis carried out on these samples, such as GC (gas chromatography) of the decomposing sample, or i.e. NMR or mass spectrometry of the recovered sample(s), to provide further confirmation of this? This would further help to address concerns around beam-induced sample damage.

Typographical, grammar, and stylistic recommendations:

1. The extent of titular alliteration is impressive, but a literature search for the term "the curious case" reveals that this is quite clichéd (>40,000 matches). Perhaps the authors could consider 'the critical case of...', or 'the crucial case of...', or similar.

2. The abstract states that "DESs incorporate renewable solids in task-specific liquids." DESs certainly can incorporate renewable solids, and certainly can be task-specific. But they are not mutually inclusive; DES can readily be prepared which are harmful, non-renewable, and for no task in particular.

3. The abstract says: "we overcame the decomposition of a quintessential DES constituent," I would argue that ChCl could be considered as *the* quintessential DES constituent.

4. The introductory statement "Unfortunately, ChCl decomposes before it melts," is quite absolute, and may require further clarification to the reader to prevent confusion, i.e. 'under typical measurement conditions.'

5. On page 2, there are two concurrent sentences which contain some repetition and could be rewritten to be less redundant; "we used (ultra-)fast differential scanning calorimetry (FDSC) to heat ChCl to temperatures where melting kinetics dominate. By heating faster than its kinetics of decomposition, we could effectively defy ChCl decomposition." Suggestion: "We defied ChCl decomposition by using (ultra-)fast differential scanning calorimetry (FDSC) to heat ChCl rapidly, to temperatures where the melting kinetics surpass degradation."

6. The labelling of Figure 2 is slightly confusing for a few reasons:

- The first part of the figure which is referenced in the text is 2C, followed by 2A & 2B.
- The labels A-E do not flow in a normal sinistrodextral way.
- In Figure 2B, or its caption, the 'direction' of the heating cycle and its effect upon the diffraction patterns could be made more obvious to the reader.

7. On page 3, I recommend changing "heating rates of 1000 K s⁻¹ on" to "heating rates of 1000 K s⁻¹ and upwards."

8. On page 3, there is a 'to' missing after 'proportional' in "directly proportional the sample mass."

9. In the first sentence of the methods, a space is missing after 'purity.'

10. "a photon flux of approximately 3x10¹² photons per second" could be changed to "a photon flux of approximately 3x10¹² s⁻¹," since it is implicit.

Response Letter to the Reviews of the Manuscript Entitled: Defying Decomposition: The Curious Case of Choline Chloride

Adriaan van den Bruinhorst^{1,*}, Jocasta Avila¹, Martin Rosenthal², Ange Pellegrino¹, Manfred Burghammer³, and Margarida Costa Gomes^{1,*}

¹École Normale Supérieure de Lyon and CNRS, Laboratoire de Chimie, Ionic Liquids Group, 46 allée d'Italie, 69364, Lyon Cedex 7, France

²Department of Chemistry, KU Leuven, Celestijnenlaan 200F, Box 2404, B-3001 Leuven, Belgium

³ESRF, The European Synchrotron, 71 Avenue des Martyrs, CS40220, 38043 Grenoble Cedex 9, France

*Corresponding authors: margarida.costa-gomes@ens-lyon.fr, adriaan.van-den-bruinhorst@ens-lyon.fr

Nature Communications,

RC: Reviewers' Comment, AR: Authors' Response, Manuscript Text

Reviewer #1

RC: *This paper describes some incredibly difficult, challenging measurements that deliver, in the end, a simple but exceptionally important results - namely, an experimentally determined measurement of the 'real' melting point and corresponding enthalpy of fusion of choline chloride. This can no be used to support theory in modelling and predicting the behaviour of choline-based deep eutectic systems based on robust experimentally determined parameters, not on extrapolated predictions. In my view, this is comparable to the ground breaking work identifying that some ionic liquids are distillable (albeit at high vacuum) that, again, enables experimental data on vapour-liquid equilibrium to be benchmarked.*

As such, this is noteworthy, significant, and should, in my view, be published.

AR: We thank the reviewer for their comments and their acknowledgement of the significance of our findings. Below we list our responses to the raised aspects in a point-by-point manner, where changes are shown as follows: additions in underlined blue, deletions in red strike-through.

Comment #1

RC: *The only area of the manuscript with which I feel a little confused or in doubt over is the extension of observations that the beta-form of choline chloride is a plastic crystal phase, and that the resilience of this to transforming back through the S-S to alpha-ChCl has, in some way, a role in formation of DES accessible across a broad range of compositions. While I think that this statement is qualitatively correct, I do not really think that this is a new concept and would highlight research in the ionic liquids field, from which DES emerged, on the utility of plastic crystal forming materials (*Phys. Chem. Chem. Phys.*, 2013,15, 1339 and *J. Mater. Chem.*, 2010,20, 2056) for consideration.*

AR: We agree with the reviewer that the concept of ionic plastic crystals as such is not new. To connect better to the ionic liquids community, we cited the most recent review from the same group as the suggested references (10.1016/j.trechm.2019.01.002). To the best of our knowledge, however, the framework of plastic crystals has not been used in DES literature to analyse the resilience to melting of common ammonium salts used in

DESs. The first quantification of the entropy of fusion of ChCl seemed to be a good occasion to emphasise the impact of plastic crystal behaviour on DES constituents and DES formation.

To further address the maturity of the ionic plastic crystal concept, we extended the discussion by comparing our results with plastic crystal behaviour in other choline salts. To support this discussion, as well as to avoid an excessive addition of new references to the manuscript, we added a table to the supporting information (Table S6) with literature data on the thermal properties of these choline salts and solid-state NMR.

~~For ions with all degrees of rotational freedom $\Delta_{\text{fus}}S_m$ can be as low as . Hence, the $\Delta_{\text{fus}}S_m$ of ChCl~~
The molar entropy of fusion of ChCl is significantly higher than that of other choline-based plastic crystals with bulkier anions ($[\text{BF}_4]^-$, $[\text{ClO}_4]^-$, and $[\text{H}_2\text{PO}_4]^-$, see Table S6). NMR studies show that the choline cation gains significant isotropy upon (or just before) the solid-solid transition. Contrary to the entropy of fusion, the entropy of transition towards the plastic crystal state is similar for the different choline salts. This indicates that the ~~asymmetric hydroxyl-functionalised choline cation retains some anisotropy in the β -ChCl crystal~~ residual entropy in the crystal is largely dictated by the anion, offering a design parameter to the enthalpy of fusion and thus the extent of the (ideal) melting point depression when forming choline-based DESs

Finally, we would like to stress that there is novelty in suggesting ionic plastic crystals as instruments to incite large melting point depressions upon mixing with a molecular compound. We advocate to develop new DESs by taking advantage of two key aspects of ionic plastic crystals:

1. The characteristically low fusion enthalpy of the pure salt
2. The favourable mixing thermodynamics observed upon adding a molecular to an ionic compound.

The second point being a result of the balance between the enthalpy (interactions) the entropy (disorder) of mixing in the liquid and fully dependent on the compatibility of mixed compounds. In conjunction with the points raised by reviewer #3, we made the changes below. Abstract:

~~Our results thus pave~~We show that DESs based on ionic plastic crystals can profit from (1) a low enthalpy of fusion and (2) favourable mixing of the way for ionic and molecular compounds. Both lower the mixture's melting point and can be altered through the nature of the ions. Ionic plastic crystal-based DESs , liquid thus offer a platform for task-specific liquids at accessible temperatures and over a broad range of compositions.

Conclusions:

Our findings for ChCl serve as a foundation for the development of DESs centred around ionic plastic crystals. ~~These mixtures~~We propose to prepare liquids from ionic plastic crystals that simultaneously profit from (1) a low $\Delta_{\text{fus}}H_m$ and ~~strong deviations from ideal eutectic behaviour, yielding liquids~~ (2) favourable mixing of the ionic and molecular compounds balancing intermolecular interactions and disorder. Both aspects can be tuned by selecting the appropriate ions and molecular compounds and significantly increase the melting point depression of the salt, yielding dense ionic fluids at accessible temperatures over a broad range of compositions. Ionic plastic ~~crystal-based~~crystal-based DESs thus provide a platform to include solids with a low environmental or safety impact in a renewable liquid solvent with task-specific properties.

Reviewer #2

RC: *The present manuscript improves the understanding of thermodynamic and morphological properties of the archetypical deep eutectic solvent (DES) component, choline chloride (ChCl), by combining fast methods. The techniques not commonly employed for studying DES and its components are fast-scanning calorimetry, synchrotron-based micro X-ray diffraction, and high-speed optical microscopy. This combination leads to a much deeper understanding of the thermodynamics of the studied components.*

The results are interesting but not groundbreaking. Besides combining the three methods, there is nothing new in the methodology.

The newly determined values for the melting point and melting enthalpy are particularly useful, but their determination leaves some open questions.

AR: We thank the reviewer for their comments. We agree that the determined fusion properties presented here are more pertinent than the methodological developments. However, we would like to highlight that it is vital to control the sample atmosphere at every stage of the measurement, which we did to an unprecedented level.

Below we list our responses to the raised aspects in a point-by-point manner, where changes are shown as follows: additions in underlined blue, deletions in red strike-through.

Comment #1

RC: *Page 4 and Fig. S16: The reported T_{fus} of 687 ± 9 K corresponds to the measured value at a heating rate of $5000 \text{ K}\cdot\text{s}^{-1}$. This value may be influenced by thermal lag and melting kinetics. For thermodynamic considerations, the value at heating rate zero should be applied. The slopes of the lines in Fig. S16 indicate an effect of about 10 K when going from $5000 \text{ K}\cdot\text{s}^{-1}$ to $0 \text{ K}\cdot\text{s}^{-1}$. This should be considered or at least discussed.*

AR: Extrapolation to 0 K/s was mentioned in the manuscript on page 5, in the same sentence where we refer to Fig. S16. While we acknowledge that an extrapolation to $0 \text{ K}\cdot\text{s}^{-1}$ —or increasing the number of measurements in general—could improve the accuracy of T_{fus} , this would not render more significant conclusions for our manuscript. We adjusted the manuscript accordingly on page 5:

~~This is also true for~~ While the accuracy of T_{fus} extrapolated could be improved by extrapolating to $0 \text{ K}\cdot\text{s}^{-1}$ from, FDSC results at different heating rates imply that this effect is small as compared to the large difference with the previously estimated T_{fus} (Fig. S16).

Comment #2

RC: *How was the onset temperature (T_{fus}) determined? Commonly, for first-order transitions, the extrapolated peak onset temperature is considered. Its determination requires at least a linear part of the leading flank of the melting peak; see, e.g., <https://doi.org/10.1007/BF01996802>. The extrapolated peak onset temperature should be available from the measured curves, and its determination does not require any peak separation procedure.*

AR: We agree with the reviewer that the onset temperature should be directly determined from the extrapolated peak onset temperature, without the need for peak separation. Following the reviewer's view, we added the values directly derived from the measured curves to the supporting information in Fig. 3, Table S1 and Fig. S16 and subsequently removed the onset temperatures derived fitted peaks. See changes in the manuscript:

... Shown are the mean molar enthalpy of fusion ($\overline{\Delta_{\text{fus}}H_m}$), ~~then mean melting point ($\overline{T_{\text{fus}}}$),~~ the model fit (orange line), ...

The mean extrapolated peak onset temperature and the corresponding standard deviation (687 ± 9 K) are the same for (1) the measured curves and the (2) single peak fitted to the curves. Hence, this value was maintained in the manuscript. This emphasises that the fitted peak represents the melting peak well.

Comment #3

RC: What is the reason for the two lines in Fig. S16?

AR: The difference between the extrapolated peak onset of (1) the fitted single melting peak and (2) the deconvoluted melting peak originates from leading flank of the decomposition peak that overlaps with the deconvoluted melting peak in the second case. Consequently, the amplitude of the deconvoluted melting peak is reduced, while the peak temperature is not. This leads to a reduced gradient in the inflection point. The extrapolation onset temperature is therefore shifted to slightly lower temperatures. Hence the difference between the two lines in Fig. S16.

This difference will not be of any importance anymore in the revised supporting information, as we followed reviewer's recommendation and derived the onset temperatures only from the measured curves.

Comment #4

RC: What is the reason for the ca. 20 K difference between the mean value of T_{fus} at 5000 K/s (687 K) and the value from the linear fits (about 665 K)?

AR: This comment is closely related to Comment #1 of reviewer 2. We believe the answer has been given by the reviewer: thermal lag and melting kinetics are the reason for the difference. An extrapolation to $0 \text{ K}\cdot\text{s}^{-1}$ could improve the accuracy of T_{fus} , but would not render more significant conclusions for our manuscript. The adjustments made to the manuscript under Comment #1 address the raised issue.

Comment #5

RC: *The reported value $T_{\text{fus}} = 687 \pm 9 \text{ K}$ seems too optimistic. It should be $(687 \pm 20) \text{ K}$ or so since there are systematic deviations.*

AR: As written in Table S2, the uncertainty of T_{fus} was calculated from the standard deviation of the sample. We took care to minimise systematic errors by changing operator, sample batch, measure at different times, and using multiple calorimetry chips for the measurements.

We agree that an extrapolation to $0 \text{ K}\cdot\text{s}^{-1}$ might improve the accuracy of T_{fus} and lead to a slightly lower value. However, the slight overestimation should not be reflected in the precision derived from the sample standard deviation. That would wrongly imply that the uncertainty also applies in the other direction to temperatures as high as 707 K. We will therefore leave the reported value (now derived from the measured curves) unchanged.

Comment #6

RC: *The value of $\Delta_{\text{fus}}H_m$ is determined at T_{fus} . This should be mentioned since the extrapolated value at $T = 298 \text{ K}$ is commonly employed for thermodynamic considerations.*

AR: We agree that we merely implied that the temperature at which $\Delta_{\text{fus}}H_m$ was the onset temperature. On page 4, we therefore added that the enthalpy of fusion was measured at the melting point and normal pressure explicitly:

The two approaches yielded similar $\Delta_{\text{fus}}H_m$ at T_{fus} and normal pressure: $13.2 \pm 4.1 \text{ kJ}\cdot\text{mol}^{-1}$ and $13.8 \pm 3.0 \text{ kJ}\cdot\text{mol}^{-1}$, respectively.

Reviewer #3

RC: *This manuscript presents results from experiments quantifying, for the first time, the fusion properties of choline chloride. In a vacuum, this may appear to be a small result but in the context of the field of Deep Eutectic Solvents (DESs), it is anything but: choline chloride typically melts before decomposing, meaning that all estimates of (non-)ideal mixing of the most common DESs are based on extrapolated data, which this manuscript essentially proves to be faulty through elegant measurements. This paper therefore represents a breakthrough and with the right framing it will interest a very wide community – not just those working in applied thermodynamics.*

In this manuscript, two primary techniques were deployed, both of which are advanced and highly experimentally challenging – namely, ultra-high-speed DSC and synchrotron X-ray diffraction. From my assessment, the work, methodology and analysis has been completed competently, and the manuscript was enjoyable to read due to the quality of writing. If my series of comments, questions and recommendations can be handled, I would unreservedly support the publication of this important article in Nature Communications.

AR: We thank the reviewer for their positive and constructive comments and we are happy they enjoyed (the presentation of) our work. Below we list our responses to the raised aspects in a point-by-point manner, where changes are shown as follows: additions in underlined blue, deletions in red strike-through.

Comment #1

RC: *Regarding the framing: I believe that the authors could do more to contextualize the broad impact of the results. At present, the introduction and conclusion focus quite strongly on the context of thermodynamic treatments of DESs. For example, in the conclusion, the authors currently state: “The accurate experimental ChCl fusion properties obtained by FDSC are imperative for a correct thermodynamic interpretation of the eutectic phase behaviour of ChCl-based mixtures,” and in the introduction, the work is described as “the long-standing issue of inaccurate ChCl fusion properties”. While these statements are indeed true and fascinating, I would argue that this work has broader impact than just the above, since it also contributes to the overall definition, and applications, of the entire field of DESs. I encourage the authors to go a step further in communicating the impacts of these results more explicitly, and accordingly strengthen introductory and concluding arguments and hypotheses such as the above.*

AR: We thank the reviewer for making us review the impact of our results to the field of DESs. We adapted the abstract and conclusions to convey this message more strongly in conjunction with the comments made by reviewer #1.

Abstract:

~~Our results thus pave~~ We show that DESs based on ionic plastic crystals can profit from (1) a low enthalpy of fusion and (2) favourable mixing of the way for ionic and molecular compounds. Both lower the mixture's melting point and can be altered through the nature of the ions. Ionic plastic crystal-based DESs, liquid thus offer a platform for task-specific liquids at accessible temperatures and over a broad range of compositions.

Conclusions:

Our findings for ChCl serve as a foundation for the development of DESs centred around ionic plastic crystals. ~~These mixtures~~ We propose to prepare liquids from ionic plastic crystals that simultaneously profit from (1) a low $\Delta_{\text{fus}}H_m$ and ~~strong deviations from ideal eutectic behaviour, yielding liquids~~ (2) favourable mixing of the ionic and molecular compounds balancing intermolecular interactions and disorder. Both aspects can be tuned by selecting the appropriate ions and molecular compounds and significantly increase the melting point depression of the salt, yielding dense ionic fluids at accessible temperatures over a broad range of compositions. Ionic plastic ~~crystal-based~~ crystal-based DESs thus provide a platform to include solids with a low environmental or safety impact in a renewable liquid solvent with task-specific properties.

Comment #2

RC: *I think that a little bit more discussion, in the context of the literature, would go a long way towards clarifying the manuscript's significance to a wider audience. The points raised throughout the manuscript about ionic plastic crystals is particularly interesting and evocative.*

Reading the manuscript implies that this is the first discovery showing that ChCl can be categorized as an ionic plastic crystal. Is this so? If yes, it could be explicitly stated. Either way, the novelty of this result requires comparison and discussion, in the context of the known plastic crystalline nature of compounds such as ChBF₄ (10.1515/zna-1997-8-923), and prior reports of the solid-solid transitions of ChCl and related quaternary ammonium compounds (10.1063/1.436673 and 10.1016/0040-6031(70)80027-2).

AR: Plastic crystal behaviour of ChCl has been anticipated before, but through the molar entropy of fusion we provide the first quantitative means to classify ChCl as plastic crystal. We explicated this on page 4:

Our results ~~classify~~ allow for the first quantitative classification of β -ChCl as an ionic plastic crystal

Following this comment and Comment #1 of Reviewer 1, we extended the discussion on plastic crystals and added a comparison with other choline-based salts and ionic plastic crystals. To support this discussion, as well as to avoid an excessive addition of new references to the manuscript, we added a table to the supporting information (Table S6) with literature data on the thermal properties of these choline salts and solid-state NMR.

~~For ions with all degrees of rotational freedom $\Delta_{\text{fus}}S_m$ can be as low as . Hence, the $\Delta_{\text{fus}}S_m$ of ChCl~~ The molar entropy of fusion of ChCl is significantly higher than that of other choline-based plastic crystals with bulkier anions ([BF₄]⁻, [ClO₄]⁻, and [H₂PO₄]⁻, see Table S6). NMR studies show that the choline cation gains significant isotropy upon (or just before) the solid-solid transition. Contrary to the entropy of fusion, the entropy of transition towards the plastic crystal state is similar for the different choline salts. This indicates that the ~~asymmetric hydroxyl-functionalised choline cation retains some anisotropy in the β -ChCl crystal~~ residual entropy in the crystal is largely dictated by the anion, offering a design parameter to the enthalpy of fusion and thus the extent of the (ideal) melting point depression when forming choline-based DESs

RC: *The authors conclude that these “findings for ChCl serve as a foundation for the development of DESs centred around ionic plastic crystals,” and write in the abstract that these “results thus pave the way for ionic plastic crystal based DESs.” The findings make me wonder whether, in fact, using ionic plastic crystals is already the*

de facto design rule for most known common DES, based on for example ChCl or quaternary ammonium salts, but that this has not yet been noticed?

AR: We agree with the reviewer that many of the ammonium salts typically used in DESs are likely to be ionic plastic crystals and that this contributes to the eutectic depth. Hence, we are currently looking into the compounds mentioned in the last paragraph. However, we would like to stress that extent of a DES constituent's melting point depression is governed by

1. the fusion properties of the excess solid and
2. the non-ideality in the liquid phase.

We would therefore not go as far as stating that a plastic ionic crystal is somehow required to form a DES, or that all plastic crystals yield DESs. To avoid possible confusion and explicate the effect of the plastic crystal nature on DES formation we added those two points to the abstract and conclusions (see changes Comment #1).

RC: *To take this argument one step further – if having some ionic plastic crystalline nature present in at least one of the components is in some way ‘key’ or ‘essential’ to a DES, such a result would be represented in experimental structure measurements. Elastic and inelastic neutron scattering data consistently shows disorder in the DES bulk, likely due to a low energy barrier to cross between different accessible structural configurations (i.e. as discussed in 10.1039/C5GC02914G, 10.1063/1.5010246 and 10.1039/C7CP01286A).*

AR: We are familiar with molecular view presented by the reviewer, which we discussed extensively in a recent perspective (ref 4, 10.1016/j.cogsc.2022.100659). However, we think that the degree of disorder in the excess solid does not directly relate to the (lack of) structure the liquid phase; the first being a pure component property and the latter being a mixture property that strongly depends on the second (or n^{th}) component.

We also think that plastic crystal behaviour is not key nor essential for DES formation. It merely is a good starting point for a large melting point depression owing to the low enthalpy of fusion.

RC: *While many other potential DES constituents decompose on or before melting, can betaine (spelt incorrectly as trymethylglycine in the concluding paragraph) be considered as such? Again, while it is not stated explicitly, the flow of the text implies this. Consider prior measurements (i.e. 10.1039/C4CP05094K).*

AR: We corrected the spelling. Betaine has repeatedly been reported to decompose upon melting (10.1016/S0022-2860(98)00613-9 and 10.1021/je500267w), complicating the determination of its fusion properties. The article the reviewer referred to presents an interesting study towards crystalline betaine, but does not seem to be relevant to above statement.

Comment #3

RC: *The authors describe visual evidence for liquefaction, gas release, and a subsequent reduction in sample size following thermal cycling. I agree that this may well be due to vaporization of ChCl, but partial decomposition is difficult to rule out without further analysis, even if the remnant ChCl recrystallizes. Was, or could, any further quantitative analysis carried out on these samples, such as GC (gas chromatography) of the decomposing sample, or i.e. NMR or mass spectrometry of the recovered sample(s), to provide further confirmation of this? This would further help to address concerns around beam-induced sample damage.*

AR: As we write on page 4, we do not rule out partial decomposition. Likely, decomposition is mainly responsible for the gas formation and reduction in sample size. We agree a more detailed study of the chemical composition

of the released gas would probably give interesting new insights into the decomposition/evaporation process. An online analysis of the gaseous products would be preferred, but is technically very challenging considering the duration of the experiment and the small sample size. We therefore believe such analysis falls outside the scope of this manuscript.

For the recovered solid samples, we think liquid chromatography + mass spectrometry would be most compatible with the sample size (few ng). However, all known thermal decomposition products are gaseous, we therefore doubt such analysis would yield sufficient new insights.

Typographical, grammar, and stylistic recommendations

RC: *The extent of titular alliteration is impressive, but a literature search for the term “the curious case” reveals that this is quite clichéd (>40,000 matches). Perhaps the authors could consider ‘the critical case of...’, or ‘the crucial case of...’, or similar.*

AR: While we are sensitive to the argument, we think that maintaining the alliteration with different words (critical, crucial) without a clear link to the work for originality sake feels a bit forced. We will therefore keep the title as is, despite it not being the most original catch-phrase.

RC: *The abstract states that “DESs incorporate renewable solids in task-specific liquids.” DESs certainly can incorporate renewable solids, and certainly can be task-specific. But they are not mutually inclusive; DES can readily be prepared which are harmful, non-renewable, and for no task in particular.*

AR: We agree with the reviewer that current statement is too absolute and does not apply to all DESs. Since we believe that novel task-specific DESs should be developed based on renewable solids, we rephrased as follows:

Through large melting point depressions, DESs ~~incorporate~~ allow for the incorporation of renewable solids in task-specific liquids ~~through large melting point depressions.~~

RC: *The abstract says: “we overcame the decomposition of a quintessential DES constituent,” I would argue that ChCl could be considered as the quintessential DES constituent.*

AR: The manuscript was adjusted accordingly:

Here we overcame the decomposition of ~~a~~ the quintessential DES constituent, choline chloride (ChCl)

RC: *The introductory statement “Unfortunately, ChCl decomposes before it melts,” is quite absolute, and may require further clarification to the reader to prevent confusion, i.e. ‘under typical measurement conditions.’*

AR: The manuscript was adjusted accordingly:

Unfortunately, ChCl decomposes before it melts when heated at typical rates ($\leq 1 \text{ K}\cdot\text{s}^{-1}$)

RC: *On page 2, there are two concurrent sentences which contain some repetition and could be rewritten to be less redundant; “we used (ultra-)fast differential scanning calorimetry (FDSC) to heat ChCl to temperatures where melting kinetics dominate. By heating faster than its kinetics of decomposition, we could effectively defy ChCl decomposition.” Suggestion: “We defied ChCl decomposition by using (ultra-)fast differential scanning calorimetry (FDSC) to heat ChCl rapidly, to temperatures where the melting kinetics surpass degradation.”*

AR: The manuscript was adjusted accordingly:

we used defied ChCl decomposition by using (ultra-)fast differential scanning calorimetry (FDSC) to heat ChCl rapidly, to temperatures where melting kinetics dominate. By heating faster than its kinetics of decomposition the melting kinetics surpass degradation (Fig. S3), we could effectively defy ChCl decomposition

- RC:** The labelling of Figure 2 is slightly confusing for a few reasons:
- The first part of the figure which is referenced in the text is 2C, followed by 2A & 2B.
 - The labels A-E do not flow in a normal sinistrodextral way.
 - In Figure 2B, or its caption, the 'direction' of the heating cycle and its effect upon the diffraction patterns could be made more obvious to the reader.

AR: Point a & b: Following both the text order and a sinistrodextral order for the figure labels would require the figure shape to become significantly less compact. We chose to privilege text order and swap subfigures A and C. Point c: We agree with the reviewer and hope that an arrow indicating the direction of time clarifies the figure.

(A) Heat flow signal of choline chloride (ChCl) obtained with fast differential scanning calorimetry at a heating rate of $5000 \text{ K}\cdot\text{s}^{-1}$ and $1000 \text{ K}\cdot\text{s}^{-1}$ (inset). The decomposition temperature (T_d) is highlighted in grey. The sample mass in the inset was determined from the integral of the ChCl solid–solid (S–S) $\alpha \rightarrow \beta$ transition and the corresponding molar enthalpy ($16.3 \text{ kJ}\cdot\text{mol}^{-1}$). (B) Temperature (T) vs. time (t) profile of choline chloride (ChCl) using a fast scanning calorimeter at a heating rate of $1000 \text{ K}\cdot\text{s}^{-1}$. The ChCl solid–solid transition temperature (T_{trs}) is highlighted in grey. (C) Corresponding XRD patterns at selected temperatures, which are highlighted as vertical grey dotted lines in the T – t profile and increase in t from bottom to top (black arrow). Also highlighted are the q of β -ChCl. (D) XRD patterns at 9 different temperatures. (E) Heat flow (Φ) signal of choline chloride (ChCl) obtained with fast differential scanning calorimetry at a heating rate of and (inset). The decomposition temperature (T_d) is highlighted in grey. The sample mass in the inset was determined from the integral of the ChCl solid–solid (S–S) $\alpha \rightarrow \beta$ transition and

~~the corresponding molar enthalpy (-)~~

RC: *On page 3, I recommend changing “heating rates of 1000 K s⁻¹ on” to “heating rates of 1000 K s⁻¹ and upwards.”*

AR: The manuscript was adjusted accordingly:

However, from heating rates of 1000 K·s⁻¹ ~~on~~ and upwards,

RC: *On page 3, there is a ‘to’ missing after ‘proportional’ in “directly proportional the sample mass.”*

AR: The manuscript was adjusted accordingly:

—both directly proportional to the sample mass—

RC: *In the first sentence of the methods, a space is missing after ‘purity.’*

AR: The manuscript was adjusted accordingly.

RC: *“a photon flux of approximately 3x10¹² photons per second” could be changed to “a photon flux of approximately 3x10¹² s⁻¹,” since it is implicit.*

AR: The manuscript was adjusted accordingly:

A monochromatic X-ray beam with a photon energy of 13 keV and a photon flux of approximately ~~3x10¹² photons per second~~ 3 × 10¹² s⁻¹

REVIEWERS' COMMENTS

Reviewer #2 (Remarks to the Author):

My concerns are sufficiently considered. I now recommend publishing the manuscript as is.

Reviewer #3 (Remarks to the Author):

Having re-read this manuscript, I am satisfied that the authors have made significant changes which have improved its rigour and impact, in response to the requests of the reviewers. I remain confident that this article is of high quality and provisionally appropriate for publication in *Nature Communications*; to continue the discussion:

1. I am satisfied with the response of the authors to my comment #1, in conjunction with the comments made by reviewer #1, which has resulted in the context of the work being conveyed more directly.
2. The discussion on plastic crystals has been extended in response to comments made by myself and other reviewers, which in my view is a positive change.
 - a. The changes made to the abstract and concluding comments help to clarify that it is not the authors' view that plastic crystals are somehow required to form a DES.
 - b. The authors argue that the degree of disorder in the excess solid does not directly relate to the liquid phase structure, because mixture properties necessarily depend on the n component. However, this seems to contradict the authors' argument that plastic crystals are a promising platform for DES. While it is true that mixture properties are more complicated than that of each single component, that does not negate the properties of the individual components. Taking the example of ChCl-Urea (as covered in the previously mentioned references), solid ChCl clearly has several viable solid phases, while urea also has different available conformations, varying between planar and pyramidal, depending on its environment; the α -ChCl and β -ChCl structures are both observed in the liquid. In other words, the mixture benefits from low $\Delta_{fus}H_m$ and favourable interplay between strong directional (H-bond) interactions and disorder, as already stated by the authors; thus, the bonding and disorder present within each n bulk solid does appear to be directly manifested in the liquid phase structure.
 - c. I am satisfied with the evidence provided in the response that betaine decomposes on melting, which was not initially fully clear to me from the manuscript.
3. I agree that the proposed studies and experiments would be significant and challenging and will not insist that this analysis should be incorporated within this manuscript.
4. I am a little confused regarding the suggestion to slightly alter the title. The authors defend their initial title in the response, but the new version of the manuscript I received appears to have been updated.
5. I am happy with the other stylistic and typographical updates, in particular the addition of the guiding arrow to Figure 2.

Response Letter to the Reviews of the Manuscript Entitled: Defying Decomposition: The Curious Case of Choline Chloride

Adriaan van den Bruinhorst^{1,*}, Jocasta Avila¹, Martin Rosenthal², Ange Pellegrino¹, Manfred Burghammer³, and Margarida Costa Gomes^{1,*}

¹École Normale Supérieure de Lyon and CNRS, Laboratoire de Chimie, Ionic Liquids Group, 46 allée d'Italie, 69364, Lyon Cedex 7, France

²Department of Chemistry, KU Leuven, Celestijnenlaan 200F, Box 2404, B-3001 Leuven, Belgium

³ESRF, The European Synchrotron, 71 Avenue des Martyrs, CS40220, 38043 Grenoble Cedex 9, France

*Corresponding authors: margarida.costa-gomes@ens-lyon.fr, adriaan.van-den-bruinhorst@ens-lyon.fr

Nature Communications,

RC: Reviewers' Comment, AR: Authors' Response, Manuscript Text

Reviewer #2

RC: *My concerns are sufficiently considered. I now recommend publishing the manuscript as is.*

AR: We thank the reviewer for their re-evaluation and are happy we could address all concerns raised in the previous revision.

Reviewer #3

RC: *Having re-read this manuscript, I am satisfied that the authors have made significant changes which have improved its rigour and impact, in response to the requests of the reviewers. I remain confident that this article is of high quality and provisionally appropriate for publication in Nature Communications; to continue the discussion:*

AR: We thank the reviewer for their thorough re-evaluation and are happy we could address the main concerns raised in the previous revision. Please find a point-by-point response to the final comments below.

Comment #1

RC: *I am satisfied with the response of the authors to my comment #1, in conjunction with the comments made by reviewer #1, which has resulted in the context of the work being conveyed more directly.*

AR: We are happy we could address the concerns raised in the previous revision.

Comment #2

RC: *The discussion on plastic crystals has been extended in response to comments made by myself and other reviewers, which in my view is a positive change.*

- a *The changes made to the abstract and concluding comments help to clarify that it is not the authors' view that plastic crystals are somehow required to form a DES.*
- b *The authors argue that the degree of disorder in the excess solid does not directly relate to the liquid phase structure, because mixture properties necessarily depend on the n^{th} component. However, this seems to contradict the authors' argument that plastic crystals are a promising platform for DES. While it is true that mixture properties are more complicated than that of each single component, that does not negate the properties of the individual components. Taking the example of ChCl-Urea (as covered in the previously mentioned references), solid ChCl clearly has several viable solid phases, while urea also has different available conformations, varying between planar and pyramidal, depending on its environment; the α -ChCl and β -ChCl structures are both observed in the liquid. In other words, the mixture benefits from low $\Delta_{\text{fus}}H_m$ and favourable interplay between strong directional (H-bond) interactions and disorder, as already stated by the authors; thus, the bonding and disorder present within each n^{th} bulk solid does appear to be directly manifested in the liquid phase structure.*
- c *I am satisfied with the evidence provided in the response that betaine decomposes on melting, which was not initially fully clear to me from the manuscript.*

AR:

- a We are happy we could clarify our view on the connection between DESs and plastic crystals.
- b We think the point raised by the reviewer is more of a paradox than a contradiction. Both the crystal structure and the local solvation environment of a substance are dictated by their functional groups. A liquid is by definition isotropic; the molecular conformation in the liquid phase might be—but is not necessarily—similar to that in the (disordered) solid, but the long-range order of a (plastic) crystal is lost upon melting. This is reflected in the contribution of the entropy of fusion to the melting point depression.
- Thermodynamically, melting can be distinguished from mixing when describing the melting point depression of a mixture. We therefore think that the contribution of plastic crystallinity to the entropy of fusion should be attributed to melting while the mixing entropy and enthalpy reflect the free energy difference between an ideal mixture of two (or more) melted solids and that of the real mixture.
- As the lack of structure of urea + ChCl illustrates, the number of possible molecular conformations of all components in the liquid mixture are determined by number of favourable and unfavourable interactions between them and the corresponding energy barriers (often called free energy landscape). These are not related to the structure in the (plastic) crystal phase. They are related, however, in the sense that at the solid–liquid equilibrium the lattice energy is similar to the energy barriers of the different liquid conformations at a given temperature.
- c We thank the reviewer for raising their doubts regarding betaine decomposition and we are happy we could clarify the manuscript.

Comment #3

RC: *I agree that the proposed studies and experiments would be significant and challenging and will not insist that this analysis should be incorporated within this manuscript.*

AR: We agree with the reviewer about the relevance of the questions raised in the previous revision and hope future work will answer these.

Comment #4

RC: *I am a little confused regarding the suggestion to slightly alter the title. The authors defend their initial title in the response, but the new version of the manuscript I received appears to have been updated.*

AR: We understand the confusion of the reviewer, as we accidentally left one of the reviewer's title suggestions in the submitted manuscript file (and that with highlighted changes). We hope the reviewer can consider this proof of us seriously considering their title suggestions. After visualisation of the alternatives, we decided to maintain the original title. In the final revisions we will change the title back to *Defying Decomposition: The Curious Case of Choline Chloride*.

Comment #5

RC: *I am happy with the other stylistic and typographical updates, in particular the addition of the guiding arrow to Figure 2.*

AR: We take this opportunity to thank the reviewer again for their detailed evaluation of the manuscript *and* supporting information. We think the quality and readability of the manuscript have improved significantly owing to their suggestions.